# A Log Ratio-Based Analysis of Individual Changes in the Composition of the Oral Microbiota in Different Dietary Phases

**DOI:** 10.3390/nu13030793

**Published:** 2021-02-28

**Authors:** Kirstin Vach, Ali Al-Ahmad, Annette Anderson, Johan Peter Woelber, Lamprini Karygianni, Annette Wittmer, Elmar Hellwig

**Affiliations:** 1Institute of Medical Biometry and Medical Statistics, Faculty of Medicine and Medical Center, University of Freiburg, Stefan-Meier-Str. 26, D-79104 Freiburg, Germany; 2Center for Dental Medicine, Department of Operative Dentistry and Periodontology, Faculty of Medicine and Medical Center, University of Freiburg, Hugstetter Straße 55, D-79106 Freiburg, Germany; ali.al-ahmad@uniklinik-freiburg.de (A.A.-A.); annette.anderson@uniklinik-freiburg.de (A.A.); johan.woelber@uniklinik-freiburg.de (J.P.W.); elmar.hellwig@uniklinik-freiburg.de (E.H.); 3Clinic for Conservative and Preventive Dentistry, Center of Dental Medicine, University of Zurich, Plattenstrasse 11, CH-8032 Zurich, Switzerland; Lamprini.Karygianni@zzm.uzh.ch; 4Institute of Medical Microbiology and Hygiene, Department of Medical Microbiology and Hygiene, Faculty of Medicine and Medical Center, University of Freiburg, Hermann-Herder-Straße 11, D-79104 Freiburg, Germany; annette.wittmer@uniklinik-freiburg.de

**Keywords:** compositional data, ratio fractions, nutrition, microbiome, oral health

## Abstract

Background: Investigating the influence of nutrition on oral health has a long scientific history. Due to recent technical advances like sequencing techniques for the oral microbiota, this topic has gained scientific interest again. A basic challenge is to understand the influence of nutrition on the oral microbiota and on the interaction between the oral bacteria, which is also statistically challenging. Methods: Log-transformed ratios of two bacteria concentrations are introduced as the basic analytic tool. The framework is illustrated by application in an experimental study exposing eleven participants to different nutrition schemes in five consecutive phases. Results: The method could be sufficiently used to analyse the interrelation between the bacteria and to identify some bacterial groups with the same as well as different reactions to additional dietary components. It was found that the strongest changes in bacterial concentrations were achieved by the additional consumption of dairy products. Conclusion: A log ratio-based analysis offers insights into the relation of different bacteria while taking specific features of compositional data into account. The presented methods allow becoming independent of the behaviour of other bacteria, which is a disadvantage of common analysis methods of compositions. The results indicate that modulations of the oral biofilm microbiota due to nutrition change can be attained.

## 1. Introduction

In recent years, the influence of nutrition on oral biofilm has received increasing attention. Dietary factors have a major influence on the microbiota of the oral biofilm and can lead to disturbances of the homeostasis of the more than 700 bacterial species within its complex network. For example, frequent consumption of simple carbohydrates can favour acidogenic and acidiuric bacterial species, consequently leading to carious lesions [1]. On the other hand, frequent consumption of dairy products, dietary fibre and certain vegetable foods is associated with lower caries incidence [2,3,4]. The calcium, phosphate and casein phosphopeptides contained in dairy products can counteract acidic demineralisation and promote remineralisation processes, thus having a caries preventive effect. In terms of plant-based foods, different ingredients are concerned with either promoting remineralisation (e.g., phosphates) or reducing bacterial adhesion and growth, thereby reducing the risk of caries development [1,5,6,7]. It is known that simple carbohydrates can lead to a higher proportion of acidogenic and aciduric species, which can promote the development of carious lesions [8]. Concerning the different bacterial species involved in the caries process, for a long time, mutans streptococci and lactobacilli were thought to be the main species responsible for the acid production and demineralisation of the tooth structure. However, in recent years it has become clear that various additional bacterial species are associated with different stages of caries development: non-mutans streptococci, such as *Streptococcus salivarius* and *Streptococcus parasanguinis*, and species of the genus *Actinomyces* have been associated with the initial stages, while *Veillonella* spp., *Lactobacillus* spp., *Atopobium* spp. and other taxa dominate the more advanced stages [9,10,11]. The effect of nutrition on the oral microbiota has mostly been investigated with either in vitro experimental approaches, testing individual ingredients, using animal experiments or analysing epidemiological data. In this paper, we consider a study analysing the influence of certain foods on the whole microbiota of the oral biofilm, was tested in situ using splint systems worn by study participants on which the oral biofilm was grown. The bacterial composition of the oral biofilm was then analysed by culture technique [1].

When analysing this type of oral biofilm data, it is common practice to examine the fractions of total concentration rather than the pure concentrations. These fractions sum up to 1, which causes special challenges in terms of the statistical analysis. Data of this kind are called compositional data, and its analysis has attracted strong attention in recent years [12,13,14,15,16]. We present a case study, in which we apply a log ratio-based approach to oral biofilm data. This approach allows analysing changes in the composition when additional standardised dietary intakes are available and the interest lies in comparing with one baseline condition (regular diet).

## 2. Materials and Methods

### 2.1. The Data

We consider a study funded by the German Research Foundation (DFG) with eleven participants and data investigating the influence of additional standardised diet components on the microbiota of the oral biofilm over 15 months. The participants underwent five phases, each of which lasted three months. During these phases, standardised dietary components in the form of sucrose, dairy products and vegetables in addition to the regular diet were investigated. Phase I was a lead-in phase without changing the nutrition habits. In phase II, participants consumed an additional 10 g of rock candy per day (Weisser Kandis; Südzucker AG, Mannheim, Germany), sucking small pieces of 2 g five times in between meals. In phase III, additional milk products were consumed (150 g of plain yoghurt 3 times daily and 100 mL of long-life milk twice a day, both 1.5% fat; Schwarzwaldmilch GmbH, Freiburg, Germany). In phase IV, 500 g of vegetable puree (types: white carrot, parsnip, carrot, “jardinière” (carrot, potato, cauliflower and pea), pumpkin and garden vegetables (carrot, potato, spinach, parsnip and leeks), Reine Weiße Karotte, Reine Pastinake, Reine Frühkarotte, and Gemüse-Allerlei (Hipp GmbH, Pfaffenhofen, Germany) and Kürbis pur and Gartengemüse (Alnatura, Darmstadt, Germany)) per day was given, before the participants returned to their individual regular diet in phase V.

An in situ splint system was used to sample the dental plaque. Individual upper jaw rigid acrylic appliances were manufactured for each study participant [17]. The splint system was taken out and stored in saline solution (0.9% NaCl) for regular meals and during oral hygiene only. It was worn while consuming the phase-specific additional foods, which were eaten slowly to expose them to the oral cavity for several minutes. During all phases, the oral biofilm was allowed to grow on embedded enamel slabs over the course of seven days. Subsequently, the splint system was removed for analysis of the oral biofilm, cleaned and after seven days it was reapplied for another seven days. This procedure was repeated three times all together. Participants brushed their teeth with standardised tooth brushes and toothpaste with a sodium fluoride content of 1450 ppm.

The oral biofilm grown in all five phases was analysed using culture techniques averaging over three measurements, which were taken within four weeks per phase. Twenty-seven groups of bacteria were identified from a total of 93 different cultivated species. Details about the course of the study and data collection can be found in [18,19].

A main focus of this work was to ascertain whether the additional standardised dietary intake leads to changes in oral microbiota. It makes sense to only consider bacteria that are usually above the detection limit. According to the results of [19], only disjunct bacterial groups were chosen, where the percentage of values above the detection limit was greater than 75%. All values below the detection limit were set to the detection limit. The bacterial groups together with the colour scheme that we use in all of our figures can be found in Figure 1. A detailed list can be found in the Appendix A and Appendix B.

If the sum of the concentrations of the bacterial groups listed here is used as the total concentration, the fractions of the single bacterial groups sum up to 1 and the data has a compositional character. For each bacterial group, participant and phase, the starting point is considered to be its fraction among the total concentration of the analysed bacterial groups.

### 2.2. Analytic Strategy

For each pair of bacteria, the ratio between the two fractions is built for each participant and phase. Such a pairwise ratio directly depicts the relation between two bacteria. They do not depend on the concentration of the other bacteria or the overall concentration. If a third bacterium has an unusually high or low concentration, this affects the overall concentration and hence the fractions, but it does not affect the ratio. Therefore, in this way we become independent of the behaviour of other bacteria. Ratios also have a simple direct interpretation describing how more frequent one bacterium is compared with the other. Note that a ratio between two fractions can also be interpreted directly as the ratio between two corresponding concentrations.The second step is to consider the logarithm with the base 10 of the ratio. This is useful because ratios tend to have a skewed and unstable distribution, whereas taking logarithms leads to a more symmetric distribution of the values. The basis 10 was chosen due to its strong interpretability: a value of 1 in the $log_{10}$ scale means the tenfold in the real concentration setting. A ratio of 1—i.e., equal concentrations—has a value of 0 in the $log_{10}$ setting. For ratios smaller than 1, the $log_{10}$ transformed value becomes negative. Therefore, a positive (negative) value means that the first bacteria is less (more) frequent than the second one.Logarithmising the ratios leads to one number per participant and phase, which can be analysed just like any other number [15]. In particular, mean values and standard deviations can be calculated. In this step, for each pair of bacteria we consider the mean value of the log ratios across all participants per phase. It is interpretable as the typical relationship between these two bacteria in this phase. Therefore, this mean value offers the opportunity to check, whether the relationship between two bacteria is similar in different phases or whether it changes due to changes in the nutrition. Accordingly, we can analyse the stability between bacteria between the phases. Similarly as above, a mean value of 1 suggests that typically the first bacterium has ten times the concentration of the second bacterium.We also analyse the corresponding standard deviations, which are a measure of the stability of the relationship between two bacteria within one phase. The closer that the value is to 0, the more homogeneous are the values across participants, i.e., the relation between the two considered bacteria is stable and can thus be assumed as “typical”. For example, if we observe a standard deviation of 0.5 and a mean log ratio of 0 for two bacteria, then the application of the 2 sigma rule leads to a 95% range of (−1, 1), i.e., 95% of the observed ratios are between 1/10 and 10.As the aim is to study the impact of an additional standardised diet on the microbiome, we also depict the changes of the concentrations compared with the first phase. For a given pair of bacteria, for each participant and phase we consider the difference in the log ratio compared with the first phase. A difference above 0 means that the ratio increased, i.e., the first bacterium became more frequent relative to the second one. As a difference in the log ratio is equal to the logarithm of the ratio of ratios, we can also interpret the values themselves. A difference of 1 means, that the ratio has increased by a factor of ten. For example, this may reflect a change from 1.2 to 12, or from 0.3 to 3. Again, we can consider mean values and standard deviations across participants per phase for phases II to V. The mean values inform us about the typical change in the relation between the two bacteria. The standard deviations offer a hint about the stability of these changes across the participants.Finally, we also consider a bacterium-specific summary measure, which is based on averaging the log ratio changes for one bacterium over all other bacteria within a single participant If this way a bacterium has a value greater than 0, this can be interpreted as having increased its fraction on average relative to all other bacteria. The bacterium with the largest positive value is the one with the most distinct increase compared with all other bacteria. We refer to this value for each participant as the average log ratio change.

A detailed mathematical description of the different quantities can be found in the Appendix B. For all means and standard deviations, 95% confidence intervals can be computed using statistical standard methods. We only report them for the most fundamental analyses, focussing on the mean values in pairs and the mean average log ratios. For the analyses, the statistics program STATA (StataCorp LT, College Station, TX, USA, Version 16.1) was used. For graphical presentation, bar and pie charts, scatter plots and forest plots were used.

### 2.3. Analysing Change or Comparing Phases Directly

In studies using a baseline condition, it is usual to consider the individual changes from the baseline, in order to investigate the impact of the subsequent conditions. However, this is not necessarily the most adequate strategy in the context of this study. It may be the case that the standardised dietary additions imply that all participants exhibit a common composition in each phase, overruling the individual differences at the baseline. In such a scenario, it would be more adequate to compare the composition directly between the conditions instead of investigating the change from the baseline. The standard deviations of the log ratios will inform us about the homogeneity of bacteria spectra in each phase, whereby we will be able to judge whether the scenario is present or not.

## 3. Results

### 3.1. A First Look at the Raw Data

In Figure 2, the fractions of the total concentrations for all ten bacterial groups are shown for each participant for each of the five phases. If the nutrition leads to a certain bacteria pattern, the pictures would have to be similar in one line each, while they would differ in the columns. This should at least apply from phase II to phase IV, as a change to a specific additional diet took place from this point onwards. However, instead of uniform patterns in each phase, we observe rather individual pattern across phases, similar to phase I. Here, the different patterns are unsurprising for the different participants, because the nutrition was in no way influenced.

If we now consider the mean values per phase (Figure 3), we see that specific bacterial groups are actually rather common in all phases. We also observe that the differences between the phases are quite small.

### 3.2. Stability of the Relation between Bacterial Groups between Phases

The relation between two bacterial groups may remain stable between the phases, or it may change. Stability may indicate that the relation is minimally affected by changes in nutrition, whereas instability may indicate sensitivity to changes. For each combination of two bacterial groups, we look at the log ratio for each participant in each phase and consider the phase-specific mean value. We can see in Figure 4 that there are differences between the pairs. At the bottom of the picture, we find pairs, where the relation to each other is very similar over all phases. At the top of the picture, we see pairs with distinct differences across the phases. Note that among the pairs with the lowest variation in the mean log ratio, we find bacterial group numbers 1, 3, 5 and 7 several times. In four instances we find bacterial group 1, and twice we find bacterial groups 3, 5 and 7. In particular, all combinations of the bacterial groups 1, 3 and 5 are among the pairs with the lowest variation in the mean log ratio.

Further, the mean log ratio in phase III is often far on the left or far on the right, i.e., in this phase we see the most extreme ratios.

### 3.3. Stability of the Relation between Bacterial Groups within Phases

The mean values above offer a hint whether the pairwise relation between bacterial groups is common in different phases. Now the question remains how stable the pairwise relation between bacterial groups is within one phase. Therefore, we take a look at the standard deviations of the phase-specific log ratios in Figure 5.

Some bacterial pairs show low values for all phases, such as 9-3, 9-10, 9-1, 5-2, 10-2 and 10-5. This means that the relation between these bacteria is stable within participants within each phase. However, even for these pairs we observe standard deviations in the magnitude of 0.5 or above, indicating rather wide 95% ranges across participants. This corresponds with the high individual variation observed in Figure 2. It is interesting to note that all six of these bacterial pairs also showed a low variation in mean values across the phases. They were all among the fourteen bacterial pairs (out of 45) with lowest range in Figure 4, i.e., these bacterial pairs tend to be stable in their relation across participants *and* phases. The opposite relation does not seem to hold: the bacterial pairs 5-3, 7-1 and 5-1 show little variation in mean values, but nevertheless high standard deviations within each phase. For some bacterial pairs, the stability substantially varies across the phases. For example, the bacterial pairs 8-1 and 7-1 show high stability in phase I and low stability in other phases, suggesting that the specific dietary supplements destroy a rather stable relation seen at the baseline. Another example is the bacterial pair 4-3, where phase III shows a much higher standard deviation than the other phases.

If we compare the phases I–IV (the nutrition in phase V was analogous to phase I by study design), the lowest standard deviation can be found in twelve out of 45 bacterial pairs in phase I, suggesting that there is no tendency towards a more uniform composition after standardised dietary supplementation. Therefore, there is no argument against analysing the changes, i.e., the changes from the baseline.

### 3.4. Identification of Bacterial Pairs with a Change in Relation in Comparison with the Baseline

In Figure 6, the mean log ratio changes with 95% confidence intervals for all pairs of bacteria are shown. While in phase II the additional diet only leads to few changes, more reactions can be observed in the other phases. The observed patterns are very similar in phases IV and V. Overall, we have to conclude that definitive statements about a “statistically significant” change in the relation in one specific phase are difficult to justify, as only few of the overall 180 confidence intervals do not include 0. However, in phase III eleven, in phase IV seven and in phase V six out of 45 confidence intervals do not include 0, i.e., we observe more than the 2.25 that we should expect under chance level. Consequently, the observed mean log ratio changes do not simply reflect random noise.

### 3.5. Stability of Change in Relation between Phases

To gain a better overview of these numbers and in particular the relation to the different phases, in Figure 7 we show the phase-specific mean values for each bacterial pair. We have sorted the pairs of bacteria by (a) their range within the different phases (small ranges on the bottom, larger ranges on the top) and (b) by the mean over the phases. In Figure 7a, we observe that the bacterial pairs 5-3, 7-1, 8-7, 5-1, 7-4, 9-1 and 9-5 show the smallest range, meaning that they show a common reaction on the additional diet in all phases, while the bacterial pairs 10-6, 9-6, 6-2, 10-8, 6-3, 6-5 and 6-1 show a large variation in their reactions in different phases. Additionally, one can see in Figure 7b that the bacterial pairs 4-2, 4-3, 4-1, 7-2, 5-2, 7-1, 10-2, 5-3 and 5-1 (8-4, 8-7, 9-4, 8-5, 9-7, 8-1, 10-4 and 9-5) show a positive (negative) value in all phases, meaning that the second bacterium tends to be more (less) frequent than the first one in each phase. Several bacterial pairs show relation changes into different directions in different phases, which could be a sign that they react differently to the nutrition change.

### 3.6. Stability of Change in Relation within Phases

The corresponding estimated standard deviations are shown in Figure 8. We observe some bacterial pairs, where the reaction to the additional diet is rather uniform across participants in all phases. These are nearly the same bacterial pairs such as in Figure 5, namely, 9-3, 7-5, 10-9, 10-5, 9-4, 9-1, 4-1 and 10-1. The bacterial pairs that always show high instability—6-2,8-2, 8-3, 6-3 and 7-3—are also nearly identical to the former observations. Note that only the pairs 9-1, 9-5, 9-4 and 8-7 with low variation in mean values across the phases also show stability within phases. By contrast, we observe lower values in standard deviations for the bacterial pair 10-1, but high variation in mean values. The least uniform reaction often occurs in phase III.

### 3.7. Interim Summary

In Table 1, we present an overview of the results of the four previous analyses: the bacterial pairs with the lowest or highest stability in the four different analyses are marked with letters differentiating between the analyses performed on the raw values (small letters) and on change values (capital letters) and between analyses within and between phases.

For example, the bacterial pair 3-1 is among the ten bacterial pairs with the highest stability with respect to the analysis of the pairwise relation between phases and the change in relation between phases, and thus it is marked as bB. The bacterial pair 4-8 is among the ten bacterial pairs with the lowest stability when analysing the pairwise bacterial relation between and within phases, and thus it is marked with bw. First of all, we can state that many pairs can be found more than once among those with the highest (or lowest) stability, indicating a rather robust classification as “stable” (or “unstable”). Sub-patterns that we typically observe are bB and wW, indicating that analysing raw values or changes provide similar results. This mainly reflects the fact that high stability in raw values also implies some stability in changes, and that low stability in raw values also makes a low stability in changes likely if the correlation between phases is only moderate.

Conceptually, it is more interesting whether stability within phases coincides with stability between phases, i.e., the bwBW pattern in the lower triangle. We can identify one bacterial pair (1-9) with high stability across all analyses. The bacterial pairs 1-5, 3-9, 5-10 and 9-10 show high stability in at least both a within-phase and between-phase analysis of the same values (bw or BW sub-pattern in the lower triangle). The relation between these pair of bacteria seems to be difficult to modify by dietary supplements or other external factors. On the other hand, we can identify two bacterial pairs with low stability in all analyses: 2-6 and 2-8 (i.e., the bwBW pattern in the upper triangle) and the bacterial pairs 3-6, 4-8 and 6-8 with low stability in at least both a within-phase and between-phase analysis (bw or BW sub-pattern in the upper triangle). The relation between these pairs of bacteria seems to be highly modifiable by diet as well as other external factors.

Pairs with high stability within phases but low stability between phases might hold the strongest interest, as these are the pairs that most distinctly react to the dietary supplements. Indeed, we can find one of such pairs 4-10 (w, W or wW in the lower triangle and b,B or bB in the upper triangle). Interestingly, there are also three pairs with the opposite pattern: 3-5, 4-7 and 7-9 (w,W or wW in the upper triangle and b, B or bB in the lower triangle), i.e., high stability between phases but low stability within phases. Finally, note that bacterial groups 6 and 8 have an entry together with nearly all other bacterial groups in the upper triangle with low stability to other bacterial groups, while the bacterial group 1 is marked with most of the bacterial groups in the lower triangle with high stability. This may reflect the notion that bacterial group 1 is rather stable and bacterial groups 6 and 8 are rather unstable. Bacterial group 10 is striking: while it still showed little variation among the participants, one has a large variation between the phases.

### 3.8. Average Log Ratio Changes

Figure 9 shows the mean values of the average log ratio changes for each bacterial group in each phase. Definite statements about a specific change in a specific phase are difficult to make, as only few confidence intervals do not include 0. However, we observe that some bacterial groups tend to increase (bacterial group 8) or decrease in frequency (4, 7) for all phases, whereas other bacterial groups tend to differ in the direction of change over phases (e.g., 6, 10). We see the smallest overall changes in phase II. In Figure 10, within each phase we sort the bacterial groups by mean values of the average log changes to focus on the phase-specific pattern in the change of the bacterial spectrum. A striking feature here is the similarity of the pattern between phases IV and V. This is quite astonishing, as in phase V the same nutrition is given as with the baseline. One possible explanation would be that the effect of the nutrition change will last longer.

#### Global *p*-Value Per Phase

When looking at the single bacterial species, the question remains whether a global effect per phase can be observed, i.e., which additional diet led to a concentration change in nearly all bacterial groups. The global *p*-value (Table 2, the test used is explained in Section B.2) only shows a value that comes near to significance (*p* = 0.055) for phase III.

### 3.9. Similarity of Change Pattern Across Bacterial Groups

Figure 6 and Figure 10 indicate some similarity in how bacterial groups react to the additional diet across different phases. We investigate this in more detail in Figure 11. We observe stronger correlations for phase II against phase V and phase IV against V. This impression is strengthened if we look in more detail at the left bottom, where all bacterial combinations are plotted.

The correlation coefficients (Table 3) confirm this impression.

## 4. Discussion

In this paper, we have presented an advantageous statistical approach to analyse a study investigating the influence of specific additional diets on the bacterial composition in the oral microbiota in an experimental manner. Traditionally, such analyses are focused on the (absolute or relative) concentration of the single bacterial species. Here, we adopt a different approach, looking at all pairs of bacteria and focusing on the quantitative relation of the two bacteria within each pair.

While this approach naturally adds complexity, it has one basic conceptual advantage that is particularly important when analysing a change in the spectrum: if the concentration or fraction of one bacterium increases or decreases distinctly, this does not directly affect the relation between any other pair of bacteria. Indeed, the simultaneous investigation of all bacterial pairs revealed that there are some pairs of bacteria that have a rather stable relation across participants within each phase. Some of them also have a rather stable relation across the phases, while some do not. Similar behaviour can be observed when examining the changes compared with the baseline.

It is rather natural that the simultaneous consideration of all bacterial pairs is not very useful to allow very specific statements for single pairs, in particular if the sample size is limited, as in our study. The general approach is mainly useful to draw an overall picture of bacterial correlation with the diet of the different phases of this longitudinal clinical study. For example, it became rather obvious that the diet of phase II had a lower impact on the composition than the dietary supplement in phases III and IV. Additionally, we could not observe the expected move back to the baseline scenario in phase V.

We could also address the fundamental question of whether an analysis of the study should focus on the individual changes in composition or on a typical composition for each nutrition. We found no evidence for the latter, and thus could stick to the intended analysis of the changes relative to the baseline.

With respect to the concrete insights gained by our analysis, the main results present the following points:The bacterial pair *Gemella Granulicatella A.* spp.–*Streptococcus* spp. group 2 is the one with high stability across all analyses.The bacterial pairs *Actiomyces.* spp.–HACEK and *Actiomyces.* spp.–*Campylobacter* spp. are the bacterial pairs with low stability across all analyses.The bacterial pair *Neisseria* spp.–*Streptococcus* spp. group 3 could be identified as the bacterial pair that most distinctly reacts to the dietary supplements.Bacterial pairs with high stability in at least both a within-phase and between-phase analysis were *Gemella Granulicatella A.* spp.–*Capnocytophaga* spp., *Rothia* spp.–*Streptococcus* spp. group 2, *Capnocytophaga* spp.–*Streptococcus* spp. group 3 and *Streptococcus* spp. group 2–*Streptococcus* spp. group 3.The most distinct changes in the spectrum were observed in phase III, i.e., the phase with additional milk products. In particular, the relation of *Neisseria* spp. to *Streptococcus* spp. group 2 and group 3 changed. When looking at the single bacterial species, the most distinct change was a decrease of *Neisseria* spp. and HACEK.In phases IV and V, when looking at the single bacterial species, the increase of *Campylobacter* spp. should be mentioned. On the other hand, the figures in phases IV and V were very similar, although in phase V the participants returned to their individual nutrition without specific additions.For *Gemella Granulicatella A.* spp., *Rothia* spp. and *Streptococcus* spp. group 3, we observed rather stable relations to other bacterial groups, while for HACEK and *Campylobacter* spp. the relations are rather variable.

At least some of these findings can be explained. As *Gemella Granulicatella A.* spp. and *Streptococcus* spp. groups 2 and 3 are kinds of the *Streptococcus* species or relatives, they can all utilise the same substrate, and consequently a similar reaction is unsurprising. This explains our main result 1 and partly the results 4 and 7. HACEK are fastidious microorganisms, which could be an explanation for the fact that they also react quickly and strongly to changing environmental conditions, which supports the results 2, 5 and 7. The high variability of *Campylobacter* spp. (result 6 and 7) could be explained by the fact that in the presence of *Actiomyces* spp. and *Streptococcs* spp. the formate can be utilised by *Campylobacter* spp., which *Actiomyces* spp. and *Streptococcus* spp. produce in glycolysis.

Note that though different microbial compositions were revealed in the different diet phases and mainly after the additional consumption of dairy products, the detected bacterial species have different potential for fermentation of carbohydrates by glycolysis leading to acids like formate, acetate and propionate. To avoid demineralisation of enamel and subsequently caries development a balanced diet is required which can sustain a balanced microbial composition of the supragingival oral biofilm.

### Further Insights

In this type of experimental setup, it cannot be expected that the individual oral biofilm returns to its original composition in a short time when participants are allowed to return to their individual nutrition without studying specific additions. Inspection of the diaries of the participants suggests that they did indeed return to the individual nutrition and did not continue to use higher levels of carbohydrates, fats and proteins. Further research including a combination of different microbiological analysing methods is necessary to understand whether this observation is indeed due to a long-lasting effect of nutrition on the oral biofilm. This would have a direct impact on the design of future studies.

In contrast to the DNA-based molecular determination technique of the microbiome, the culture technique depicts active and viable bacteria [20]. In a comprehensive analysis of the microbiota of endodontic infections, the authors of the aforementioned study even showed that some bacterial species, that could be isolated and identified by the culture technique could not be detected by the culture-independent cloning technique. On the other hand, it has been shown that up to 50% of the oral bacteria could not be detected by the culture technique [21]. Zaura et al. [22] showed that healthy individuals exposed to a single antibiotic treatment undergo considerable microbial shifts of the microbiome of their faeces, while their salivary microbiome composition remains more robust and stable. The minor changes in bacterial composition revealed in the present study deliver another hint regarding the stability of the oral microbiota, including in the supragingival oral biofilm.

## 5. Conclusions

Systematically considering the log ratio of concentrations between bacterial pairs has offered new insights into the influence of nutrition on the oral biofilm and the relation of specific bacterial groups with each other. The methods make it possible to circumvent some typical problems encountered in the analysis of compositional data.

## Figures and Tables

**Figure 1 nutrients-13-00793-f001:**
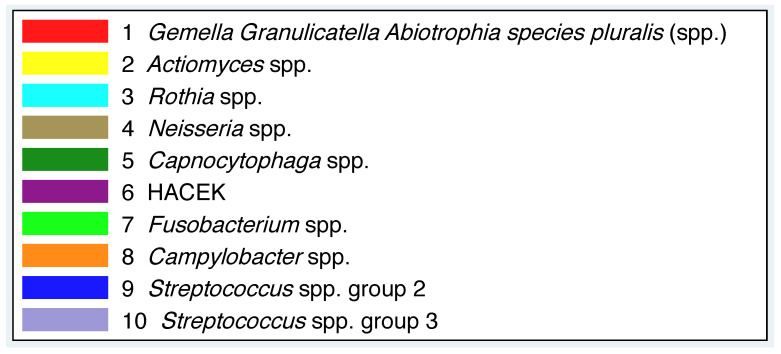
Bacterial groups and colour scheme for all figures.

**Figure 2 nutrients-13-00793-f002:**
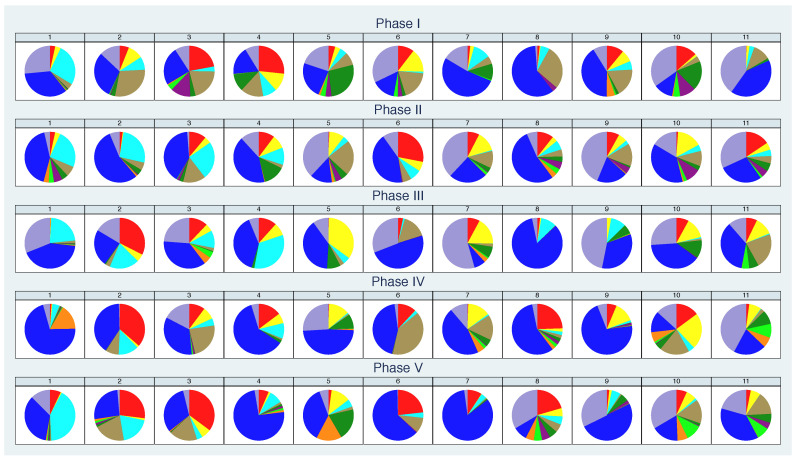
Fractions of the bacterial groups on the total concentration per participant (column) and phase (row) based on mean values per phase.

**Figure 3 nutrients-13-00793-f003:**
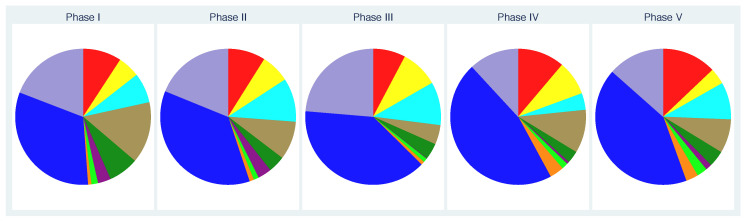
Fractions of the bacterial groups on the total concentration per phase based on mean values per phase.

**Figure 4 nutrients-13-00793-f004:**
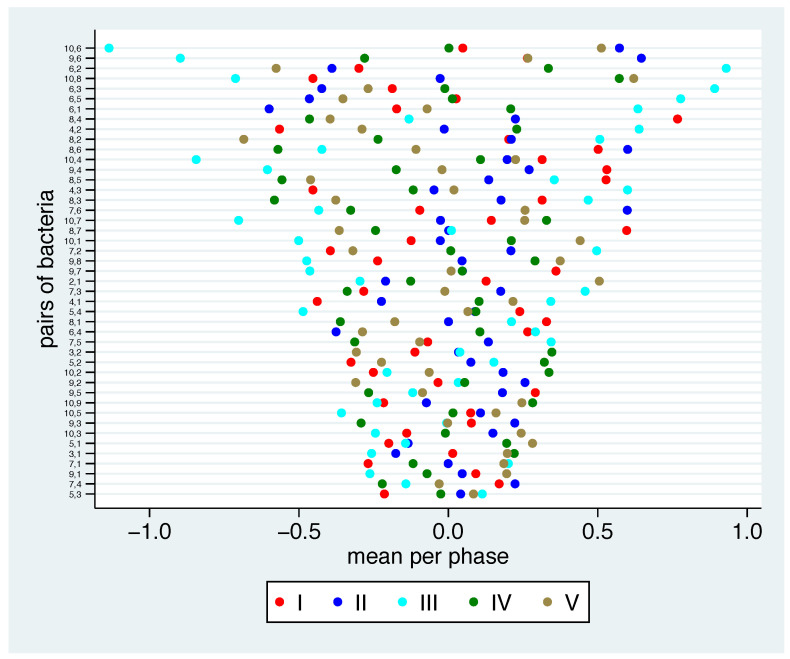
Mean log ratios for all bacterial pairs of bacterial groups and phases. The values are centred around 0 for each bacterial pair. The bacterial pairs are ordered according to the range of the mean values.

**Figure 5 nutrients-13-00793-f005:**
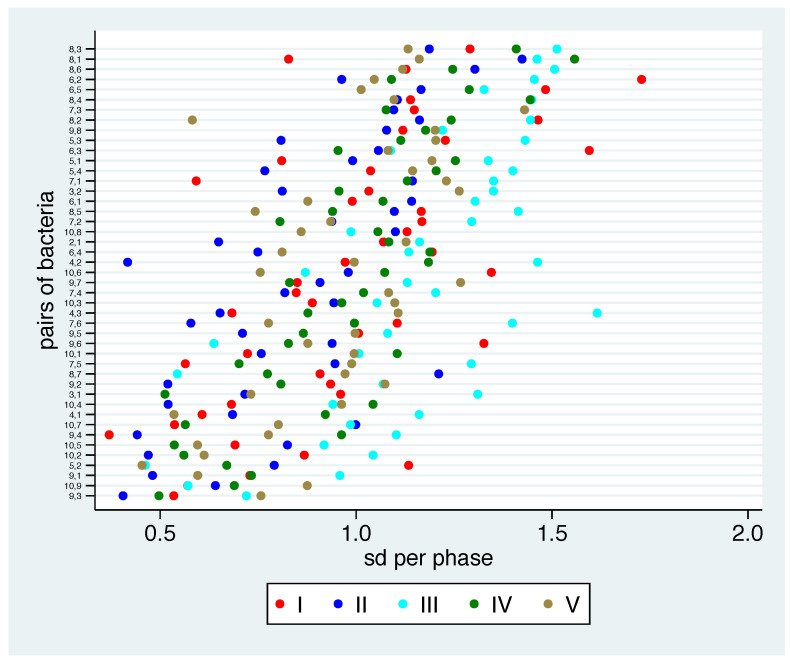
Standard deviations (sd) for phase-specific log ratios for all bacterial pairs of bacterial groups and phases based on empirical estimates. The pairs are sorted by the mean sd value over all phases.

**Figure 6 nutrients-13-00793-f006:**
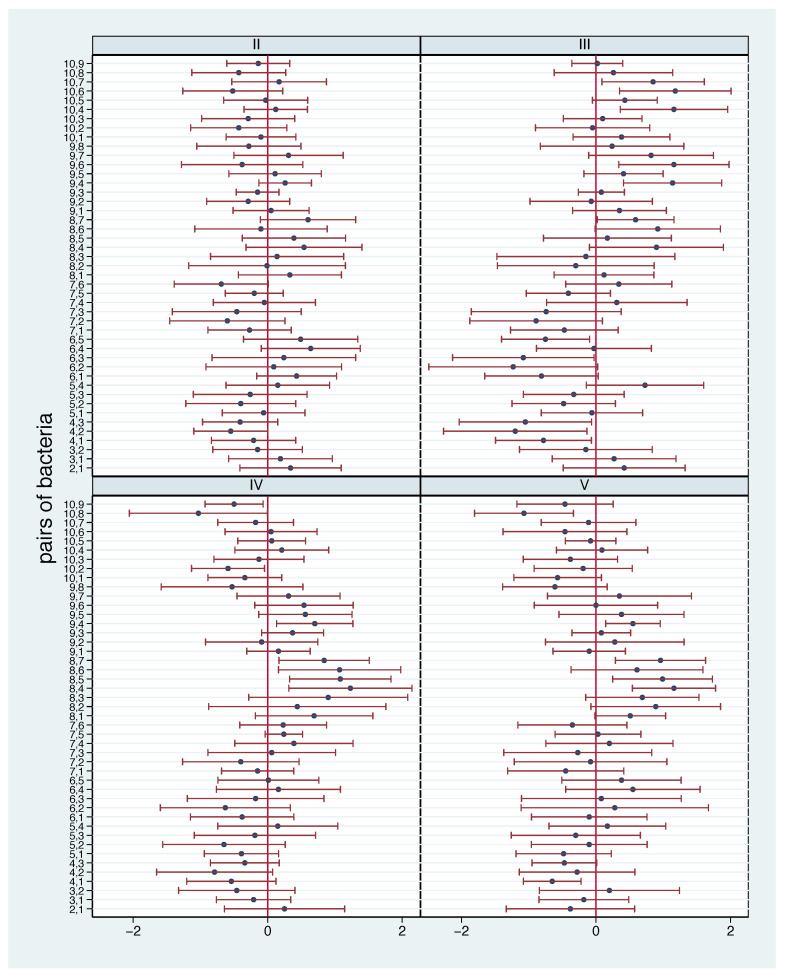
Mean log ratio changes for all bacterial pairs for phases II–V with 95% confidence intervals.

**Figure 7 nutrients-13-00793-f007:**
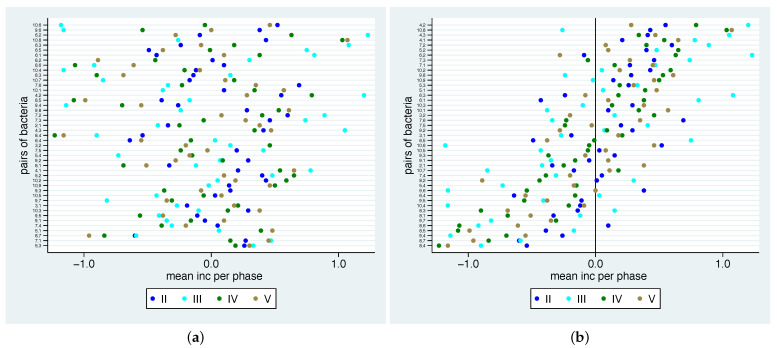
Mean values for phase-based log ratios for changes for all pairs of bacteria and phases sorted by range (**a**) and mean (**b**).

**Figure 8 nutrients-13-00793-f008:**
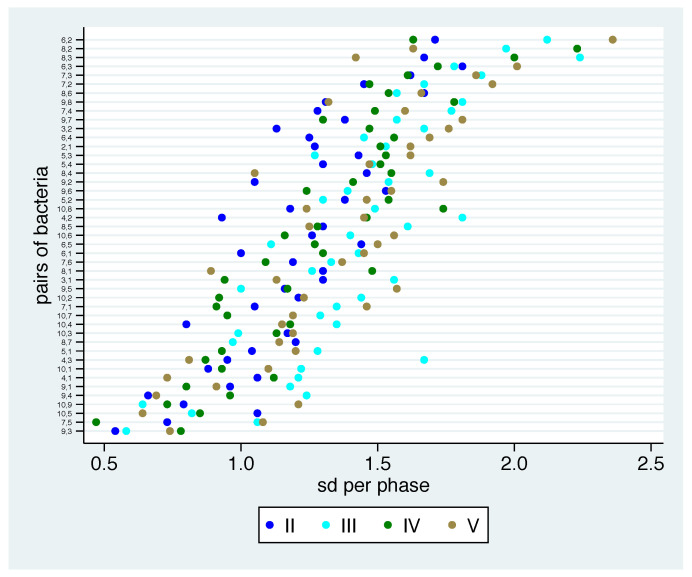
Standard deviation (sd) for phase-based log ratios for changes for all pairs of bacteria and phases estimated empirical and sorted by mean of sd over the phases.

**Figure 9 nutrients-13-00793-f009:**
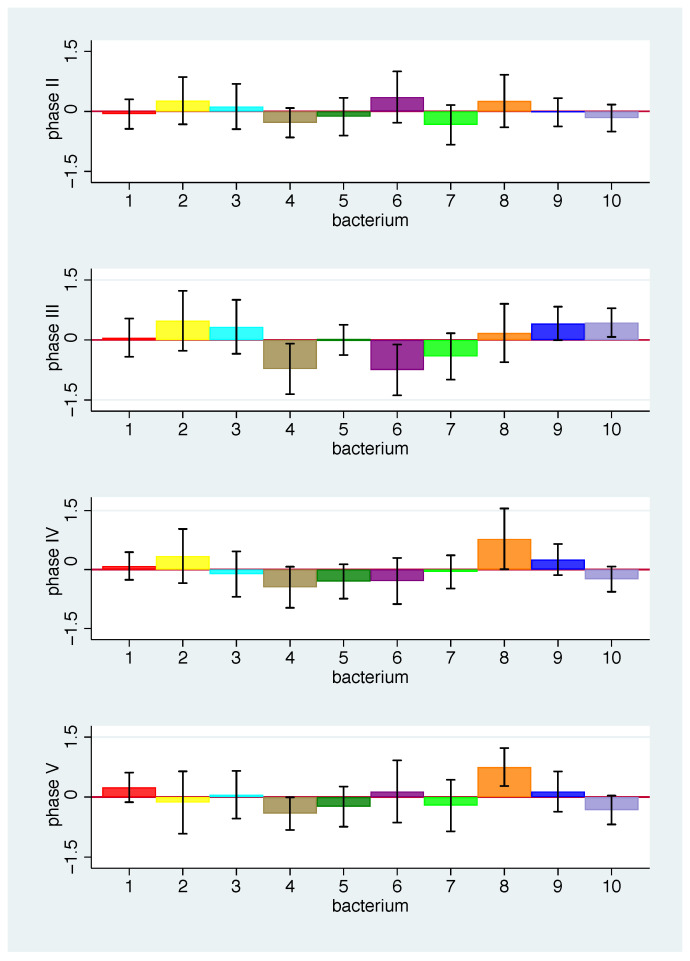
Mean values of average log ratio changes with 95% CI for bacterial groups 1 to 10 for phase II (**top**) to phase V (**bottom**). For colour scheme, see Figure 1.

**Figure 10 nutrients-13-00793-f010:**
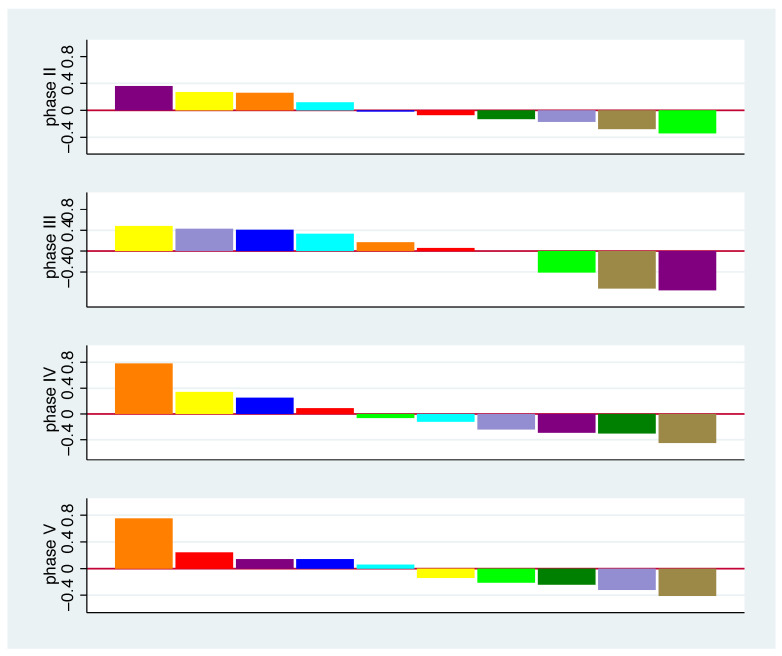
Mean values of average log ratio changes sorted for bacterial groups 1 to 10 for phase II (**top**) to phase V (**bottom**). For colour scheme, see Figure 1.

**Figure 11 nutrients-13-00793-f011:**
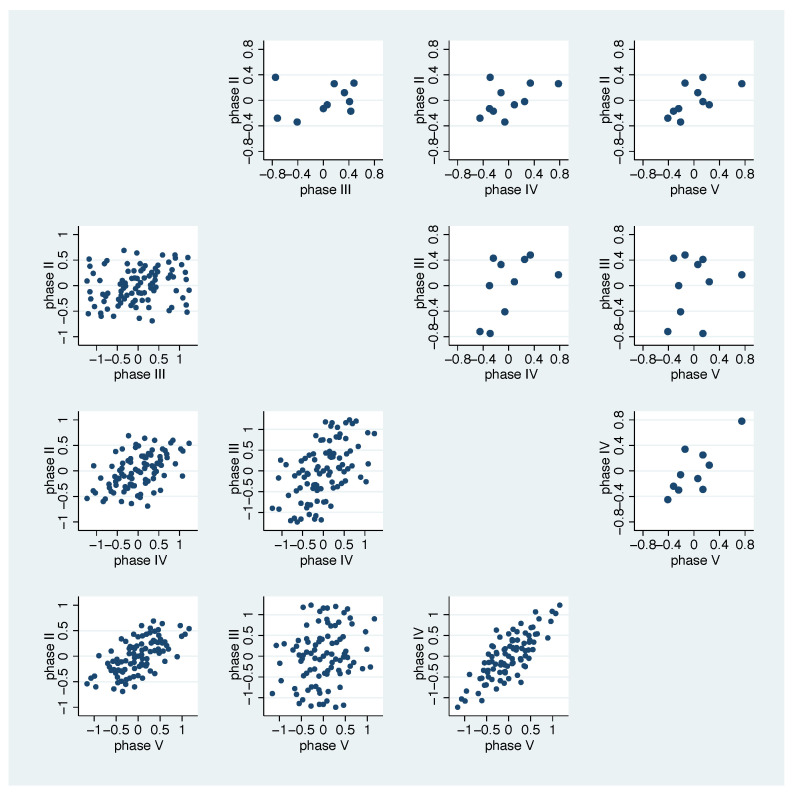
Scatter plots of the mean values of bacterial groups’ specific average log ratio changes. Half-matrix right top: mean value for each bacterial group in one phase against the other phases. Half-matrix left bottom: mean value for each bacterial pair in one phase against the other phases.

**Table 1 nutrients-13-00793-t001:** Overview of results from the analyses in Section 3.2, Section 3.3, Section 3.4, Section 3.5 and Section 3.6: The ten bacterial pairs with the highest (half-matrix left bottom) and the lowest (half-matrix right top) stability in each analysis are marked with b (between) and w (within) phases and corresponding capital letters for changes in comparison to baseline.

	*1*	*2*	*3*	*4*	*5*	*6*	*7*	*8*	*9*	*10*
1						*bB*		*w*		
2				*b*		*bwBW*	*W*	*bwBW*		
3	bB				*w*	*bBW*	*wW*	*wW*		
4	wW		W				*W*	*bw*		*B*
5	bBW	w	bB	W		*bwB*				
6								*wBW*	*bB*	*bB*
7	bB			bB	W				*W*	
8							B		*wW*	*bB*
9	bwBW		bwW	w	B		B			
10	W	w	bB	w	bwW		w		bwW	

**Table 2 nutrients-13-00793-t002:** Global *p*-value per phase.

	Phase II	Phase III	Phase IV	Phase V
*p*-value	0.459	0.055	0.096	0.289

**Table 3 nutrients-13-00793-t003:** Correlation coefficients for bacterial groups’ specific average log ratio changes between phases.

	Phase III	Phase IV	Phase V
**Phase II**	0.199	0.488	0.628
**Phase III**		0.528	0.205
**Phase IV**			0.780

## Data Availability

The data are available on request from the authors.

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
