# Peer review of "A Log Ratio-Based Analysis of Individual Changes in the Composition of the Oral Microbiota in Different Dietary Phases"

_nutrients, 2021, doi:10.3390/nu13030793_

Round 1

Reviewer 1 Report

[Suggestions]

L. 312-332: 
The referee would like to read the discussion on the main results (#1 - #7), just like the nutritional description of (L. 333-339), such as the substrate, formate, and glycolysis etc. 

References #1: should be "Appl Environ Microbiol 86: e01421-20".

L. 22: "frequent requent consumption" should read "frequent consumption".

Author Response

Point-to-point answer to review 1:

Dear Reviewer,

Thank you for carefully reading the manuscript. Your response to the single points are marked in red.

  1. 312-332: 
    The referee would like to read the discussion on the main results (#1 - #7), just like the nutritional description of (L. 333-339), such as the substrate, formate, and glycolysis etc. 

Response: Thank you for this proposal. A point-by-point explanation is somewhat difficult, as some of the explanations are relevant to several main results. Nevertheless, we have now shown which of the listed points provides explanations for which result.

We have changed the Lines 333-339 to:

At least some of these findings can be explained.

As Gemella Granulicatella A. spp and  Streptococcus spp. groups  2 and 3  are kinds of the   Streptococcus species  or relatives, they can all utilise the same substrate,   and consequently a similar reaction is unsurprising. This explains our main result 1 and partly the results 4 and 7.

HACEK are fastidious germs, which could be an explanation for the fact that they   also react quickly and strongly to changing environmental conditions, which supports the results 2, 5 and 7.

The high variability of  Campylobacter spp. (result 6 and 7) could be explained by the fact that in the presence of Actiomyces spp. and Streptococcs spp.  the formate can be utilised by  Campylobacter spp., which Actiomyces spp. and Streptococcus spp. produce in glycolysis.

Additionally we added:

It should be kept in mind that though different microbial compositions were revealed in the different diet phases and mainly after the additional consumption of dairy products, the detected bacterial species have different potential for fermentation of carbohydrates by glycolysis leading to acids like formate, acetate and propionate. To avoid demineralization of enamel and subsequently caries development a balanced diet is required which can sustain a balanced microbial composition of the supragingival oral biofilm.

References #1: should be "Appl Environ Microbiol 86: e01421-20".

Response: Thank you for this point. We changed the reference and added 86, e01421-20.

  1. 22: "frequent requent consumption" should read "frequent consumption".

Response: We changed this accordingly.

Reviewer 2 Report

This  study consists of an in depth statistical approach to analyse the influence of specific diets on the bacterial profile of the oral
microbiota. Log-transformed ratios of two bacteria concentrations are introduced as the  basic analytic tool in this study.In this vein, a  study exposing  participants to different nutrition schemes in five consecutive phases was studied and evaluated.The authors studied globally all pairs of bacteria and focused on the quantitative relation of the two bacteria within each pair.Results showed that  some pairs of bacteria  have a  stable relation across participants within each phase or across the phases, while other not.

The study is well designed and statistical analysis evaluated the different correlations in depth.

However, I think that the authors must mention if their study in humans was approved by any ethical committee.

The log ratio-based analysis showed interesting results into the relation
of different bacteria. Important shiftings were showed in the oral microbiota when compared with the baseline.

The paper is well written.

Author Response

Dear Reviewer,

Thank you for carefully reading the manuscript. Your response to the single points are marked in red.

However, I think that the authors must mention if their study in humans was approved by any ethical committee.

Response: Thank you for this hint. We actually forgot to provide this information.

We added at the end of the manuscript:

Ethics approval and consent to participate:

The study protocol was approved by the Ethics Committee of the University of Freiburg (Nr. 237/14). A written informed consent was obtained from all participants.

All experiments and data collections were performed in accordance with relevant guidelines and regulations.
